# Effect of Different Probiotic Fermentations on the Quality of Plant-Based Hempseed Fermented Milk

**DOI:** 10.3390/foods13244076

**Published:** 2024-12-17

**Authors:** Yingjun Zhou, Yifan Xu, Shuai Song, Sha Zhan, Xiaochun Li, Haixuan Wang, Zuohua Zhu, Li Yan, Yuande Peng, Chunliang Xie

**Affiliations:** 1Institute of Bast Fiber Crops, Chinese Academy of Agricultural Sciences, Changsha 410205, China; zhouyingjun@caas.cn (Y.Z.); xyf18894359005@163.com (Y.X.); zhansha@caas.cn (S.Z.); li-xiao-chun@foxmail.com (X.L.); zhuzuohua@126.com (Z.Z.); yanli214@126.com (L.Y.); ibfcpyd313@126.com (Y.P.); 2Jiangsu Junyao Life Technology Development Co., Ltd., Yancheng 224100, China

**Keywords:** hempseed fermented milk, probiotics, volatile flavor compounds, non-volatile flavor substances, correlation analysis

## Abstract

This study investigated the effects of three different single-strain probiotics *Lactiplantibacillus plantarum* XD117, *Lacticaseibacillus paracasei* LC-37, and *Lacticaseibacillus rhamnosus* LGG, on the quality of hempseed fermented milk. The main findings were that adding probiotics increased the inhibition rate of α-glucosidase and pancreatic lipase in hempseed fermented milk significantly. Non-targeted metabolomic correlation analysis results confirmed that 14 substances, including three flavonoids, six amino acids and their derivatives, and five short peptides, were positively correlated with the hypoglycemic and hypolipidemic activities of hempseed fermented milk. Furthermore, a total of 59 volatile flavor compounds were identified, including aldehydes, alcohols, ketones, acids, and esters, and the role mapping of different probiotic communities was provided. These results can guide the development of hempseed fermented milk with unique flavor, rich probiotic content, and significant functional characteristics.

## 1. Introduction

Fermented milk has been used as human food for a long time. It is a dairy product with a unique flavor made by fermenting fresh milk as a raw material. It has an acidic flavor and is easy to solidify. It has a huge market worldwide because it is more nutritious and has a richer flavor than fresh milk. Traditional fermented milk is usually made by inoculating lactic acid bacteria (LAB) for fermentation [1]. Consumers are attracted not only by its slightly sour and creamy taste but also by its associated health benefits. However, fermented milk may cause lactose intolerance, animal protein allergy, and cardiovascular disease caused by high saturated fat content in some individuals, which limits the consumption of fermented milk worldwide. In addition, the amount of animal protein is insufficient to meet the daily needs of humans in developing countries. Based on these, plant-based fermented milk as a non-dairy substitute has become a preferred choice for consumers and a research hotspot in the food and beverage industry [2].

Hempseeds are derived from ripe and dried hemp fruit (*Cannabis Sativa* L.) and have been an edible food ingredient for thousands of years [3]. They are rich in nutrients, including 25 to 35% oil, 20 to 30% carbohydrates, 20 to 25% protein, 10 to 15% insoluble fiber, and vitamins and minerals such as potassium, magnesium, phosphorus, calcium, sulfur, iron, and zinc [4]. Of these, 85% of the fatty acids in hempseeds are polyunsaturated, particularly the ratio of linoleic acid (ω-6) and α-linolenic acid (ω-3) was between 2:1 and 3:1, which is the ideal ratio for optimal health and has been effectively developed and utilized. The remaining portion after oil extraction (also known as defatted hemp/hemp cake/hemp meal) is typically abandoned as a byproduct or used as an animal feed with little economic value. However, it still contains many nutrients such as proteins, dietary fiber, etc. [5]. Wasting this portion results in a great waste of resources; paying attention to how these byproducts can be processed into products with significant nutritional value is worthwhile. 11S globulin and 2S albumin are the main proteins in hempseeds, containing all the essential amino acids (with good digestibility) in the content and proportions required for human dietary needs [6]. It is worth mentioning that arginine in hemp protein accounts for 12%, which is the highest content of all plant seed proteins [7]. Arginine, an endogenous amino acid, is involved in the synthesis of various products responsible for regulatory functions in the body [8], with a special application value in the development of functional food. High protein content and such nutritional properties made defatted hemp an ideal raw material for plant-based fermented milk preparation. Hence, defatted hemp has received high attention as a raw material in the development of functional fermented milk in recent years. Xu et al. [9] altered the composition of gut microbiota in loperamide-induced constipated rats and successfully increased fecal quantity, fecal water content, and small intestine transport rate, reduced inflammatory damage, and effectively prevented constipation by feeding fermented fermented milk supplemented with 10% defatted hemp. The large amounts of acetate, propionate, and butyrate produced during the fermentation process of hemp seed beverages are beneficial for the growth of probiotics and have strong prebiotic activity [10]. In fermented hemp milk containing kefir grains, there was an increase in the content of nutrients such as organic acids, free fatty acids, and amino acids, as well as in the type and concentrations of flavor compounds such as alcohols, acids, and esters. Compared to unfermented products, the texture of the final product, including particle size, viscosity, and resistance to phase separation, was also increased [11]. The above results confirmed the potential application of hemp seed in the preparation of functional fermented milk.

Probiotics are living microorganisms that have many beneficial functions to the human body, such as regulating glucose and lipid metabolism, enhancing immunity, relieving constipation, and reducing inflammation. They have a wide application in fermented milk products, and it was found that they could improve the flavor and enhance the beneficial functions of fermented milk. However, numerous studies have shown that unique aromas and compounds depend on the probiotics used [12]. *L. rhamnosus*, *L. paracasei*, and *L. plantarum* are commonly used probiotics in fermented milk, which have been successfully applied in soy fermented milk [13,14]. In this study, we successfully prepared pure plant-based hempseed fermented milk with defatted hemp as raw material first. Next, we systematically analyzed the effects of three commonly used probiotics on the quality of fermented milk, such as total acid, viable bacteria count, hypoglycemic activity, texture, hypoglycemic activity, and flavor substances. Finally, the mechanisms of these influences were elucidated through metabolomics analysis, providing an impact map of different probiotic communities. Our findings will guide the development of hemp-fermented milk with distinct flavors and high nutritional value.

## 2. Materials and Methods

### 2.1. Microbial Culture and Preparation

*L. paracei* LC-37 and *L. rhamnosus* LGG were isolated from commercially available fermented milk products. *Streptococcus thermophilus* and *Lactobacillus delbrueckii* subsp. *bulgaricus* are derived from yogurt starter Mid1.0 by the supplier (Chr. Hansen, Hørsholm, Denmark). *L. plantarum* XD117 was screened out by our lab and stored in the China Center For Type Culture Collection (CCTCC) with accession numbers CCTCC M 20241730. The LC-37, LGG, and XD117 strains were placed under anaerobic conditions at 37 °C and cultured for 24 h in de Man, Rogosa, and Sharpe agar liquid medium. Later, these cells were collected by centrifugation at 6000 rpm for 10 min and stored at 4 °C, then transferred to phosphate buffer saline (PBS buffer, 0.02% KH_2_PO_4_, 0.115% Na_2_HPO_4_, 1% tryptone, 0.8% NaCl and 0.1% sodium glutamate).

### 2.2. Preparation of Hempseed Fermented Milk

Three strains of probiotics were inoculated in MRS Broth medium for activation. Then, the bacterial concentration was adjusted to 1 × 10^9^ CFU/mL (OD_600_ ≈ 1.0), centrifuged at 4 °C at 6000 rpm for 5 min, washed three times with sterile PBS buffer (pH 7.2), and recycled as the seed liquid for fermentation. Fresh defatted hempseed was washed with 0.1% sodium carbonate solution to clean the impurities, then mixed with clean water (1:9, *w*/*w*) and homogenized using a colloid grinder (JML-65, Qiangzhong Machinery Technology Co., Wenzhou, China) at 2900 r/min for 10 min. The resulting suspension filtered with 200 mesh sandbags was supplemented with 0.25% soybean peptide, 0.8% stabilizer, and 8% sucrose. The mixture was stirred and homogenized (D500, Dalongxingchuang experimental instrument Co., Beijing, China) for 10 min, then pasteurized for 10 min at 90 °C and rapidly cooled to 42 °C. Fermented milk was divided into five groups: group HMU (unfermented hempseed milk), group HMD (0.04% fermented milk starter Mid1.0), HMP (0.04% fermented milk starter Mid1.0 + 0.5% *L. plantarum* XD117), HMC (0.04% fermented milk starter Mid1.0 + 0.5% *L. paracasei* LC-37) and HMR (0.04% fermented milk starter Mid1.0 + 0.5% *L. rhamnosus* LGG). Four types of LAB seed solutions were inoculated into a mixture of hemp milk and fermented at 42 °C. Hempseed milk samples were collected during the fermentation period (0–8 h) and post-ripening period (12 h) and placed in an ultra-low temperature freezer at −80 °C for testing.

### 2.3. Viability and Acid Production Analysis

The number of viable bacteria in hempseed fermented milk was determined every 2 h using the method described by Jia et al. [15]. The hemp seed fermented milk was divided into 50 mL centrifuge tubes for fermentation treatment in incubators. Serial dilutions were made by adding 1 mL of fresh fermented milk to 9 mL of 0.1% physiological saline (from 10^−1^ up to 10^−5^), and 0.1mL of the diluent was evenly spread on MRS medium. The culture medium was cultivated at 37 °C for 48 h. A plate with a colony counts between 30 and 300 was selected for counting. The enumeration of LAB was conducted in three repetitions.

The titratable acidity (TA) of five groups of fermented milk (HMU, HMD, HMP, HMC, and HMR) were measured during the fermentation and post-ripening periods, and the effect of probiotics and fermentation agents on acid production capacity was determined in combination. 10 g of fermented milk sample and 20 mL of aseptic water were mixed thoroughly in an aseptic sampling bag. Then, TA in fermented milk with NaOH standard solution (0.05 mol/L) was titrated with the titration endpoint at pH 8.2 measured using a pH meter.

### 2.4. Texture

The texture of fermented milk was determined by a texture characteristic analyzer (TA.XTC-18, Baosheng Industrial Development Co., Ltd., Shanghai, China). Firstly, a 40 mm TA/BE anti-extrusion probe was selected. Afterward, the pre-measurement, measurement, and post-measurement rates were set to 0.5 mm/s, 0.2 mm/s, and 0.2 mm/s, respectively. The compression degree was 20 mm, and the measured temperature was 25 °C. The following indicators were measured: the hardness, cohesiveness, adhesiveness, consistency, and chewiness of fermented milk.

### 2.5. Functional Analysis (α-Glucosidase and Pancreatic Lipase Inhibition)

The in vitro α-glucosidase inhibitory activity test was performed spectrophotometrically. In this assay, a centrifuge tube was prepared with 1 mL of α-glucosidase (1 × 10^−3^ mg/mL) and 0.2 mL of sample. The resulting solution was incubated at 37 °C for 10 min. Subsequently, 0.5 mL of p-nitrophenyl—α-Dglucopyranoside (p-NPG, 5 mM) was added to the centrifuge tube, and the reaction was allowed to proceed for another 10 min at 37 °C. Finally, the reaction was terminated by adding 1 mL of Na_2_CO_3_ (0.1 M) solution. The inhibition rate of α-glucosidase was calculated based on the absorbance of the reaction mixture measured at 405 nm using a microplate reader (SpectroMax M2, San Jose, CA, USA).

The inhibitory activity of pancreatic lipase was evaluated as follows. Briefly, 50 µL PBS (pH 7.4), 50 µL pancreatic lipase solution (5 mg/mL), and 50 µL diluted sample solution were mixed together and incubated in the water bath at 37 °C for 10 min. Subsequently, 50 µL of p-nitrophenyl palmitate (p-NPB, 11.2 mM) was added to the above mixture to start the reaction and incubated at 37 °C for 20 min, then placed in an ice bath for 2 min. Finally, the mixture was transferred to a 96-well microplate, and the inhibition rate of pancreatic lipase was calculated based on the absorbance measured at 405 nm using the microplate spectrophotometer.
(1)$$\mathrm{The}\;\mathrm{inhibition}\;\mathrm{ratio}=1-\frac{A4-A3}{A2-A1}\times 100\%,$$
where A_4_, A_3_, A_2_, and A_1_ are the ODs of the tested group, the blank-tested group, the control group, and the blank control group, respectively.

### 2.6. Volatile and Non-Volatile Compound Analysis

According to the published method, IMS instruments equipped with automatic sampling devices (FlavourSpec^®^, Dortmund, Germany) were used in this study. Further analysis and modification were performed using an Agilent 490 gas chromatograph (Agilent Technologies, Palo Alto, CA, USA)—equipped with an MXT-5 capillary column (15 m × 0.53 mm ID, 1 µm FT) from Dortmund, Germany—to test volatile compounds in the samples. In short, 1.0 g of folded fermented milk sample was placed into a 20 mL headspace vial and incubated at 50 °C for 15 min. Next, 300 µL of headspace gas was automatically injected into the syringe using a heated syringe at 85 °C (500 rpm; no diversion mode). Volatile compounds were separated using an MXT-5 capillary column (isothermal at 50 °C), with nitrogen gas (99.999% purity) as the carrier gas and a program flow rate of 2 mL/min for 2 min. The flow rate should be increased to 100 mL/min within 10 min and then to 150 mL/min after 10 min until the flow stops. Volatile compounds were identified by comparing RI and drift time with those in the GC-IMS library and NIST database. The result was expressed as the average relative content of each volatile compound to the total aroma content. Chemometric analysis was performed on the obtained IMS data using instrument software VOCal 0.4.03, which included plugins such as Laboratory Analysis Viewer (LAV), GC-IMS library search, and reporter and library drawing plugins.

10 mL of the sample was centrifuged at 4 °C (4000× *g*) for 10 min. The supernatant was collected and filtered through a 0.22 μm nylon filter. All samples were mixed as a quality control (QC) sample. Metabolic spectra were analyzed by LC separation and Q-TOF (Q-Exactive quadrupole orbital trap mass spectrometer, Thermo Fisher Scientific, Waltham, MA, USA) using ESI positive and negative ion modes. C18 chromatographic columns (150 mm, 2.1 mm i.d., 1.6 mm, Phenomenex, Sartrouville, France) were used for chromatographic separation. The mobile phase was composed of 0.1% formic acid (A) and acetonitrile containing 0.1% formic acid. Use the following gradient: 0 min, 5% B; 2 min, 5% B; 4 min, 25% B; 8 min, 50% B; 10 min, 80% B; 14 min, 100% B; 15 min, 100% B; 15.1 min, 5% and 16 min, 5% B separated metabolites. The parameters of positive and negative ionization mode and electric spray ionization source were as follows: the mass range of *m*/*z* is 100–1200. The resolutions of the full MS scan and HCD MS/MS scan were 70,000 and 17,500, respectively. LC-MS/MS data processing and multivariate analysis were performed as described. Differential expressions of metabolites in the experimental group were identified using orthogonal partial least squares discriminant analysis (OPLS-DA). The statistical standard for selecting characteristic metabolites with variable importance (VIP) in OPLS-DA prediction was considered to be differentially expressed if the *t*-test is higher than 1.0 and the q-value is less than 0.05.

### 2.7. Statistical Analysis

All experiments were conducted in triplicate. The differences between samples can be analyzed using SPSS software for one-way ANOVA and Duncan’s multiple range test. Origin 2018 (OriginLab, Northampton, MA, USA) was used to draw parcels. A *p*-value < 0.05 is considered statistically significant. OmicStudio tools, such as PCA and heat maps, were used for bioinformatics analysis.

## 3. Results and Discussion

### 3.1. Effect of Different Probiotics on TA and Viable Counts of LAB in Hempseed Fermented Milk

The total acid production of different probiotics was determined by measuring the total acid (Figure 1A), and the measurement results were expressed as lactic acid equivalents (*w*/*w*). Before fermentation, the TA in hempseed milk was low (only 0.15%). As the fermentation proceeded, carbohydrates were used by fermenting organisms to produce acid, mostly lactic acid, but also short-chain fatty acids, which lower the pH of the milk [16]. During the initial 4 h of fermentation time, the TA of hempseed milk increased rapidly from 0.16% to 1.13–1.18%, and the total acid stabilized over the next 4 h. This result was similar to the previous report in soy fermented milk inoculated with Danisco mixed probiotics, where acidity increases rapidly during the first 3 h of fermentation [17]. However, the differences between groups gradually increased due to the addition of probiotics. The most significant effect on fermented milk TA was the addition of the probiotic *L. rhamnosus* LGG, which reached 1.29% in the latter part of the 12-h ripening period. Moreover, the TA of hempseed fermented milk added with *L. plantarum* XD117 and *L. paracasei* LC-7 was higher than that of hemp milk fermented only using the fermented milk starter mid1.0. This result indicates that the probiotic *L. rhamnosus* LGG has a higher acid-producing capacity, indicating it might have a different fermentation property compared to the other two types of LAB fermentation [18].

Fermented milk containing sufficient amounts of probiotic LAB can provide health benefits to the human body [19]. As shown in Figure 1B, during fermentation, the growth and vitality of the starters *S. thermophilus* and *L. bulgaricus* were almost unaffected by the probiotics added to the fermentation system. However, during the storage period, the HMD group without added probiotics had the highest live bacteria count, and the number of viable counts was higher than that of the probiotic groups (HMR, HMC, and HMP). This may be due to the opposition of fermented milk starters with probiotics to nutrients during storage. The addition of probiotic LAB decreases the pH value of fermented milk, which affects the surface charge of the bacterial cell membrane and reduces bacterial stability [20,21].

### 3.2. Effect of Different Probiotics on the Texture of Hempseed Fermented Milk

Texture is an important quality indicator of fermented milk that directly affects consumer acceptance [22]. The results of texture profile analysis (TPA) of different fermented milk samples showed that the supplementation of probiotics could significantly affect the parameters of hardness, adhesiveness, and resilience (*p*  <  0.05) (Table 1). Hardness is an important indicator for evaluating the texture of fermented milk; the greater the hardness, the better the clotting. Table 1 shows that the lowest hardness value was observed for hempseed fermented milk fermented with *L. plantarum* XD117, with a 2.5% reduction in hardness compared to HMD. Previous studies have confirmed a close correlation between the hardness of fermented milk and its protein content [23]. Therefore, the reduced hardness in this group (HMP) may be related to protein metabolism.

Adhesiveness is also a key characteristic because the thickness of fermented milk is higher and has greater adhesion. Comparatively, there was a decrease in the adhesiveness of HMR, HMC, and HMP by 6.52%, 5.07%, and 5.6%, respectively (Table 1). During the fermentation process of fermented milk, the slime produced by LAB is mainly composed of extracellular polysaccharides such as xylose, galactose, arabinose, mannose, etc., which increase the viscosity of fermented milk [24]. The addition of probiotics may lead to reduced adhesiveness, thus affecting the metabolism of the fermented milk starter.

There is a positive correlation between resilience and elasticity of fermented milk [25]. In this study, we found that the addition of *L. paracasei* and *L. plantarum* had no significant effect on the springiness and resilience of hempseed fermented milk. However, the resilience of hempseed milk fermented with *L. rhamnosus* LGG was lower compared to the control group HMD. Awad et al. [26] reported that the hydrolysis of para κ-casein through the action of the starter cultures reduces the springiness. Therefore, the decreased resilience of fermented milk in the HMR group may be related to the hydrolysis of hemp proteins by *L. rhamnosus* LGG. In general, the addition of probiotics had little effect on the texture of hempseed fermented milk.

### 3.3. Inhibition Assay of Hemp Seed Fermented Milk on α-Glucosidase and Pancreatic Lipase

Fermented milk is a nutrient-dense food that may be beneficial for individuals suffering from chronic diseases such as hypertension, diabetes, hyperlipidemia, and cardiovascular diseases [27]. However, due to the different bioactive compounds generated by different fermentation agents, there is a need to screen LAB-fermented milk for its anti-diabetic and anti-hyperlipidemic potential [28]. In this study, the inhibitory activities of α-glucosidase and pancreatic lipase in different probiotic-fermented hempseed milk are displayed in Figure 1C and Figure 1D, respectively. In the α-glucosidase inhibition assay, hempseed fermented milk produced using *L. plantarum* XD117 (95.9 ± 1.27%), *L. rhamnosus* LGG (88.33 ± 0.96%), and *L. paracasei* LC-7 (31.39 ± 0.89%) exhibited significantly higher inhibitory activities than that produced with the fermented milk starter Mid1.0. Among them, fermentation with *L. plantarum* XD117 had the most significant enhancement effect on the α-glucosidase inhibition rate in hempseed fermented milk, which is in accordance with the results reported in soybean fermented milk [28]. A similar result was found in the pancreatic lipase inhibition assay, where pancreatic lipase inhibition rates in hempseed fermented milk with the addition of the three different probiotics were obviously higher than those of the fermented milk using fermented starters alone. In comparison, the addition of probiotic *L. plantarum* XD117 exhibited the strongest inhibitory effect on pancreatic lipase (92.58 ± 1.31%) in hempseed fermented milk. Numerous studies have demonstrated the health benefits of fermented soymilk supplemented with probiotics in lowering blood glucose levels and improving lipid profile in diabetic/hyperlipidemic mice [29,30,31,32,33,34,35]. However, to our knowledge, this is the first report demonstrating that probiotic fermentation can enhance the potential anti-diabetic and anti-hyperlipidemic effects of hempseed milk.

Nakashima et al. [28] reported that soy fermented milk produced using *L. plantarum* TOKAl 17 had the highest inhibition rates of α-glucosidase and was significantly related to the accumulation of flavonoids after fermentation. Flavonoids are natural secondary metabolites with polyphenolic structure and low molecular weight that are widely present in plants. According to numerous in vitro and in vivo pharmacological studies, flavonoids have a potentially positive effect in treating diabetes and hyperlipidemia through multiple pathways and targets [36]. Flavonoids are also one of the important bioactive components in hempseed. Pojic et al. [37] analyzed the phenolic profile of hempseed and identified 27 kinds of flavonoids, including apigenin, eriodictyol, genistein, luteolin, naringenin, naringin, quercetin, and rutin. All of them have been reported to have anti-diabetic and anti-hyperlipidemic activities [38]. However, these flavonoids are usually tightly bound to insoluble polymers through hydrophobic interactions, hydrogen bonding, and covalent bonds, with their bioactivities and bioavailability being largely limited [39]. Previous studies have shown that LAB can produce glucosidase, which releases bounded flavonoids through enzymatic hydrolysis, thereby improving their biological activity and bioavailability [40]. Therefore, we speculate that the enhancement in α-glucosidase and pancreatic lipase inhibitory activities of probiotic-fermented hempseed fermented milk may be related to the transformation and release of flavonoids.

In addition, peptides derived from the hydrolysis of plant proteins exhibit a wide range of biological activities both in vivo and in vitro, such as anti-hypertensive, antimicrobial, antioxidant, hypocholesterolemic, and immunomodulatory. In the last decade, a number of peptides with serum glucose-regulating and hypocholesterolemic effects have been purified from hempseed hydrolysates [41,42,43,44]. Many strains of LAB not only convert fermentable carbohydrates into lactic acid but also possess protein hydrolysis activity, which is crucial for many fermentation processes [45]. Therefore, it can be speculated that the formation of peptides may be one of the key reasons for the improvement in hypoglycemic and hypolipidemic activities of hempseed milk after probiotic fermentation. However, the peptide profiles produced by different probiotic fermentation still need to be further analyzed.

### 3.4. Volatile Compounds in Hemp Seed Fermented Milk Samples as Identified via GC-IMS

Gas chromatography-ion mobility spectrometry (GC-IMS) is characterized by high separation efficiency, rapid response, and high sensitivity. Therefore, GC-IMS can be used to detect volatile flavor compounds in four types of hempseed fermented milk samples. Migration time is an effective parameter for the mass and geometric structure of ions in a reactive chemical substance and is determined by the normalization of the ion migration time and the peak value of the active ions [46]. In Figure 2, each column displays the signal peak of the same volatile organic compounds in different samples, while each row shows all the enriched volatile substances in the same sample. The horizontal axis represents the selected feature recognition peaks, and each column corresponds to one VFC. The vertical axis is the sample identification number, and each row represents one sample. Black and blue are the background colors, with higher and lower contents indicated by red and light colors, respectively. Grouping substances with similar patterns of change facilitates observation and comparison. These groups were then categorized into five regions: A, B, C, D, and E.

According to the results of GC-IMS testing, the characteristic peak intensity of the aroma substances was found to be positively correlated with their contents in the sample. A total of 59 volatile flavor compounds were identified in the four hempseed fermented milk samples, including 19 aldehydes, 15 alcohols, 11 ketones, 3 alkenes, 3 esters, 2 carboxylic acids, 2 sulfides, 1 furan, and 3 other volatile metabolites. Among these substances, aldehydes, alcohols, ketones, terpenes, and esters were the main volatile substances in the four samples. Appendix A analyzes the relative content of different substances, and we identified 19 important flavor compounds with relative content >1% in the four samples (Table 2). These compounds had an enhancing effect on the flavor of the product.

In order to improve the characterization of the different volatile compounds, the relative differences of the aromatic compounds in the four samples were calculated based on the signal strength on the fingerprint spectrum, as shown in Figure 3. The similarity of the obtained VFC index can be evaluated using statistical methods such as heat maps and clustering, as shown in Figure 3. Volatile flavorings were analyzed and vertically clustered for the three samples, as displayed in Figure 3B. There were several differences in the relative characteristics of volatile classification among HMD, HMP, HMR, and HMC. It is clear from the graph that the primary ingredients of hempseed fermented milk are HMR (containing 27.42% aldehydes, 25.67% ketones, 22.02% alcohols, 12.84% carboxylic acids, and 8.98% esters), HMP (containing 18.56% aldehydes, 26.35% ketones, 26.96% alcohols, 17.94% carboxylic acids, and 7.23% esters), followed by HMC (containing 15.85% aldehydes, 24.8% ketones, 25.92% alcohols, 19.76% carboxylic acids, and 9.69% esters), and HMD (containing 15.89% aldehydes, 23.2% ketones, 29.51% alcohols, 16.76% carboxylic acids, and 9.62% esters).

These products contained the highest content of acetic acid and ethanol, and the threshold for this compound was relatively low, playing a decisive role in the overall flavor formation process. Previous studies have found that acetic acid and ethanol are potent flavor compounds in fermented milk fermentation [47,48], which is in agreement with our findings. In addition, five significant alcoholic flavor compounds were observed, and their relative contents changed differently after probiotic fermentation. For instance, the relative contents of 1-hexanol, 1-pentanol, and 1-pentene-3-alcohol increased significantly after fermentation with *L. plantarum*, resulting in a fresh, tropical, fruity, winey, sweet, and green odor. In contrast, the relative content of the other four alcoholic flavor compounds, namely 1-hexanol, 1-pentene-3-alcohol, 1-butanol, and 1-propanol, were significantly decreased in the hempseed fermented milk with the addition of *L. rhamnosus*, with the exception of 1-pentanol. These changes also resulted in a more harmonious and rich flavor of the fermented milk.

Compared to HMD, the proportion of ketones in probiotic-fermented hempseed milk was significantly increased. During the fermentation process, microbial metabolism produces ketones, which are mainly produced by the oxidation of esters and unsaturated fatty acids [49]. Many of these components have a gratifying aroma, which may explain the relative decrease in ester levels in probiotic-fermented hempseed milk. Among these important ketone volatile flavor compounds, 3-hydroxy-2-butanone and 2, 3-butanedione caught our attention. Previous studies have reported that 2,3-butanedione plays an important role in the formation of fermented milk flavor by providing mainly the aroma of buttery or butterscotch [50,51], while 2,3-butanedione can generate 3-hydroxy-2-butanone via diacetyl reductase, which also imparts cultured, buttery, and sweet aromas to fermented milk [52]. In this study, the contents of both 2,3-butanedione and 3-hydroxy-2-butanone were found to be considerably higher in *L. plantarum*-fermented hempseed milk, while a significant increase was observed in *L. paracasei*-fermented hempseed milk, suggesting that the addition of *L. plantarum* may be more beneficial for the formation of the typical flavor of fermented milk. In addition, the main secondary metabolites produced by the self-oxidation of unsaturated fatty acids are aldehydes, which can give plant-based hempseed milk its unique flavor. According to the information in Table 2, the relative content of aldehydes was higher than that of HMD in the HMP and HMR groups of fermented milk, indicating that probiotic fermentation enriches the pleasing flavor in hempseed milk.

Principal Component Analysis (PCA) reduces the dimensionality of a dataset composed of many related variables while preserving as much variation as possible, making it a multivariate statistical method. This method has been widely used to study the changes in food aroma compounds. Figure 4 shows the PCA results of the peak intensities of volatile substances in the four samples. The contribution rate of the first principal component (PC1) was 41%, while that of the second principal component (PC2) was 29%. The combined contribution rate of the first and second principal components was 70%, containing most of the information on the four samples and presenting the main characteristics of the volatile aroma. The four sample components showed clear separation in the principal component analysis chart, indicating that the GC-IMS technology can be applied to determine volatile substances in the samples. To better distinguish the samples, GC-IMS can be combined with PCA. Meanwhile, the research results showed that there were significant differences among the four samples. In conclusion, the choice of probiotic organisms in hempseed fermented milk has an important influence on the production of flavor substances, but the mechanism of these differences remains to be further explored.

### 3.5. Non-Volatile Flavor Substances

From the above-mentioned results, it can be seen that probiotics have a significant influence on the quality of hempseed fermented milk. Accordingly, a metabolomics study was conducted to explain and explore the effects of probiotics on non-volatile metabolites in hempseed fermented milk from a metabolic perspective. In the PCA scoring chart, the larger the difference between the samples, the farther the distance between the two samples [53]. As illustrated in Figure 5A, clustering between samples from the same group demonstrated good sample reproducibility and data reliability. The further distance between the HMD, HMP, HMR, and HMC groups indicated higher variability in the non-volatile metabolites of the four types of fermented milk. The four hempseed fermented milk were distinguished in accordance with PCA1 (18.9%) and PCA2 (13.1%).

The OPLS-DA scoring plot shown in Figure 5B indicates a stronger degree of clustering differentiation between the HMD, HMP, HMR, and HMC groups. To prove the reliability of the OPLS-DA model, the sample sequence was randomly arranged. When establishing the OPLS-DA model, the variable order defining the classification Y was randomly arranged 200 times. The applicability and predictability of the model were evaluated using R^2^ and Q^2^, respectively. As can be seen from the models, the slopes of the R^2^ and Q^2^ regression lines were both >0, and the intercepts of the Q^2^ regression line were both <0 (Figure 5C), indicating a successful validation of the models. In addition, Figure 5F visualizes the potential labeled compounds that contributed to the differences between the HMD, HMP, HMR, and HMC groups. The differential metabolites between HMD, HMP, HMR, and HMC were statistically analyzed using screened variable importance in projections (VIP) > 1, |*p* (1)| > 0.05, |*p*_corr_ (1)| > 0.5, and an S-plot was established to display covariance (*p*) and correlation (*p*_corr_). In the S-plot, the greater the metabolic difference between the two groups, the longer the distance from the midpoint of the selected variable.

When screening for differential metabolites, it is generally believed that *p*-values ≤ 0.05, VIP ≥ 1, and fold changes > 1.5 or ≤ 0.75 are statistically significant. A total of 247 differential metabolites were acquired and initially recognized by comparing mass to MS/MS fragmentation, charge ratio (*m*/*z*), and precise mass and mass spectrometry fragmentation patterns guided by the Human Metabolome Database. These 247 differential metabolites included 41 amino acids and peptides, 25 lipids, 19 carbohydrates, 11 flavonoids, 7 diterpenoids, 7 carbonyl compounds, 7 benzoic acids, and their derivatives, and other compounds (Figure 6A). A hierarchical cluster was constructed to visualize the similarity of all hempseed fermented milk samples, where each column represents one sample (Figure 6B). The more similar the metabolic profiles of the samples, the more likely they were to be grouped together in the tree. Figure 6B clearly shows the variations between four hempseed milk groups fermented with different probiotics.

To identify the metabolic pathways of metabolites differentially expressed in the four types of hempseed fermented milk, the MetaboAnalyst analysis platform was selected with reference to the Kyoto Encyclopedia of Genes and Genomes (KEGG) database. KEGG compounds were identified using the Pathway Library, and the reference model organism *Arabidopsis thaliana*; compound name matches were manually checked using KEGG or HMDB. Based on the assay results (Figure 7), it is evident that fermentation with probiotics has a great influence on the metabolic pathways of hempseed milk. Compared to the HMD group, 17 pathways were significantly affected in the hempseed fermented milk supplemented with the probiotic *L. plantarum* XD117, involving amino acid biosynthesis and metabolism, unsaturated fatty acid biosynthesis, carbohydrate metabolism, nucleotide metabolism, biosynthesis of cofactors, and biosynthesis of secondary metabolites. In *L. rhamnosus* LGG fermented hempseed milk, 17 metabolic pathways were also significantly affected, including amino acid biosynthesis and metabolism, biotin metabolism, sulfur relay system, phosphotransferase system, galactose metabolism, nucleotide metabolism, biosynthesis of cofactors, and biosynthesis of secondary metabolites. The number of metabolic pathways significantly affected in *L. paracasei* LC-37-fermented hempseed milk was much more than that of HMP and HMR. In addition to the above-mentioned metabolic pathways, other pathways related to antibiotic synthesis, vitamin B6 metabolism, and linoleic acid metabolism were also significantly affected. The differences in these metabolic pathways may also be an important reason for the varying biological efficacy of hempseed milk fermented with different probiotics.

To demonstrate the relationship between the differences in metabolites of flavonoids, amino acids, and peptides in the four hempseed fermented milk groups and their contribution to pancreatic lipase and α-glucosidase inhibitory activities, a Pearson’s correlation coefficient (r) analysis was completed, and the results were presented in Table 3. Table 3 clearly shows that 14 potential markers were positively correlated with α-glucosidase inhibitory activity or pancreatic lipase inhibitory activity (*r* > 0.6), and the correlations are statistically significant (*p* < 0.05). Flavonoids are the most important class of plant secondary metabolites present in all hemp plant organs, including seeds. It was previously reported that microbial fermentation is conducive to the release of conjugated flavonoids and the conversion of flavonoid glycosides to aglycones [40,54]. Meanwhile, flavonoid aglycones typically exhibit stronger biological activities than flavonoid glycosides due to the release of more hydroxyl groups after glycosidic bond cleavage [50]. In this study, we found that the relative contents of all three flavonoids, panasenoside, genistein, and rutin, increased significantly after fermentation with probiotics and showed a better positive correlation with the indicator related to α-glucosidase inhibition (*r* > 0.90). This result is consistent with previous research findings on the excellent α-glucosidase inhibitory activities of these three flavonoids [55,56]. Therefore, the release and transformation of panasenoside, genistein, and rutin during fermentation may be an important reason for the enhanced hypoglycemic activity of probiotic-fermented hempseed milk.

In fermented plant protein beverages, protein hydrolysis produces a variety of bioactive short peptides, amino acids, and their derivatives. These amino acids and peptides are important components of fermented milk and have a wide range of biological activities, such as risk of thrombosis, lowering of hypertension, and lowering of cholesterol, as well as providing useful antioxidant effects [57]. A total of six amino acids, their derivatives, and five short peptides were positively correlated with α-glucosidase inhibitory activity or pancreatic lipase inhibitory activity in four hempseed fermented milk groups. Among them, the compounds that significantly correlated with α-glucosidase inhibitory activity were L-arginine, L-lysopine, N_2_-gamma-glutamylglutamine, and glutaminylmethionine, while the compounds that significantly correlated with pancreatic lipase inhibitory activity were L-arginine, L-tyrosine, N-methyl-L-glutamic acid, N-acetyl-L-phenylalanine, 4-guanidinobutanoic acid, glycylleucine, hydroxyprolyl-lysine, and tryptophylhydroxyproline. Of these, L-arginine is the only compound that has been shown to be significantly correlated with both glucosidase inhibition and pancreatic lipase inhibitory activity in hempseed milk. L-arginine, as a cationic semi-essential amino acid, can be involved in many physiological processes. It has been shown to act as a precursor to nitric oxide by relaxing blood vessels, effectively promoting wound healing, improving cardiovascular disease, and stimulating growth hormone secretion [58]. Numerous small clinical and experimental studies have shown that L-arginine can correct endothelial dysfunction related to hyperlipidemia, hypertension, coronary artery disease, smoking, diabetes, and obesity via enteral or parenteral administration [58,59]. This strongly supports our results that L-arginine is significantly related to the inhibition of α-glucosidase and pancreatic lipase in hempseed milk. The results of this study also demonstrated that probiotic fermentation greatly increased the contents of L-arginine in hempseed milk by 3.66–5.42 times. Besides, there were few studies that reported the regulation effects on blood lipid and blood sugar of the other five amino acids and short peptides. Therefore, it is necessary to explore their functional activities in future studies.

## 4. Conclusions

This study investigated the effect of a single probiotic on the quality of hempseed fermented milk. The addition of probiotics had little effect on titratable acidity, viable bacteria count, and textural properties but significantly increased the inhibition rate of α-glucosidase and pancreatic lipase. GC-IMS conducted a general and validated analysis of the main flavor components, and a total of 19 important flavor compounds (with a relative content higher than 1%) were found to be the key factors in distinguishing the different aroma characteristics of hempseed milk fermented with four different probiotics. Based on the results of LC-MS analysis, seven compounds were screened by correlation analysis for significant positive correlation with the hypoglycemic activities of the hempseed milk, including three flavonoids (panasenoside, genistein, and rutin), two amino acids (L-arginine and L-lysopine) and two short peptides (N2-gamma-glutamylglutamine and glutaminylmethionine). Seven of the compounds showed a significant positive correlation with hypolipidemic activities, including five amino acids (L-arginine, L-tyrosine, N-methyl-L-glutamic acid, N-acetyl-L-phenylalanine, and 4-guanidinobutanoic acid) and three short peptides (glycylleucine, hydroxyprolyl-lysine, and tryptophylhydroxyproline). This work provides a detailed understanding of the effect of various probiotics on volatile and non-volatile metabolomic characteristics of hempseed milk and may provide a reference for the development of functional plant-based fermented milk.

## Figures and Tables

**Figure 1 foods-13-04076-f001:**
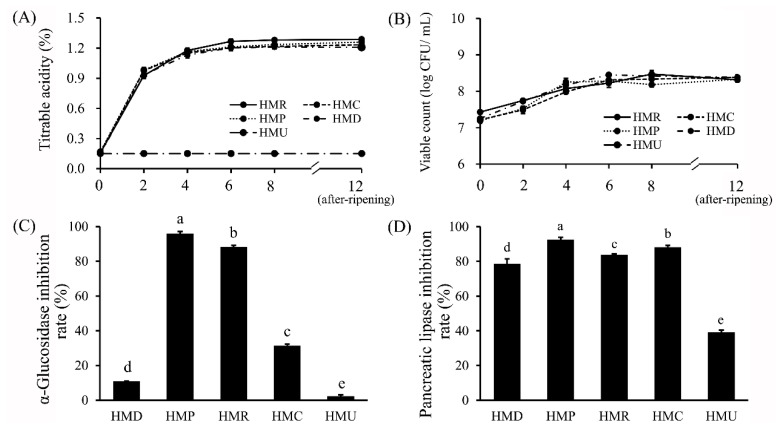
Effect of different probiotics on viable bacteria count, TA, α-glucosidase inhibitory, and pancreatic lipase inhibitory capacity of hempseed milk before and after fermentation. (**A**) Effect on the TA of hempseed fermented milk. (**B**) Effect on the viable count in hempseed fermented milk. (**C**) Effect on fermented milk α-glucosidase inhibitory capacity. (**D**) Effect on pancreatic lipase inhibitory of fermented milk. Different letters above group columns indicate significant differences (*p* < 0.05).

**Figure 2 foods-13-04076-f002:**
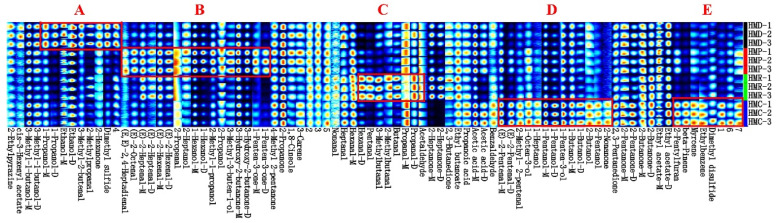
Fingerprints of volatile components in hempseed milk fermented with different probiotics. Dark and light colors represent the change in the relative concentration of the substance from high to low; -M and -D represent the monomeric and dimeric forms of the same substance, respectively. These differential compounds on the fingerprint are mainly concentrated in these 5 regions (**A**–**E**).

**Figure 3 foods-13-04076-f003:**
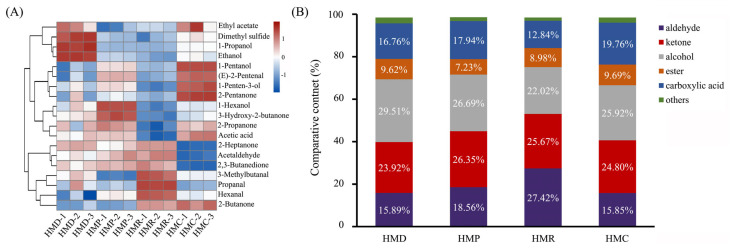
Cluster heat map (**A**) and difference map (**B**) of relative content of volatile flavor compounds in different hempseed fermented milk. Note: in the clustered heatmap, blue stands for relatively low levels and red for relatively high levels, with darker colors representing larger differences.

**Figure 4 foods-13-04076-f004:**
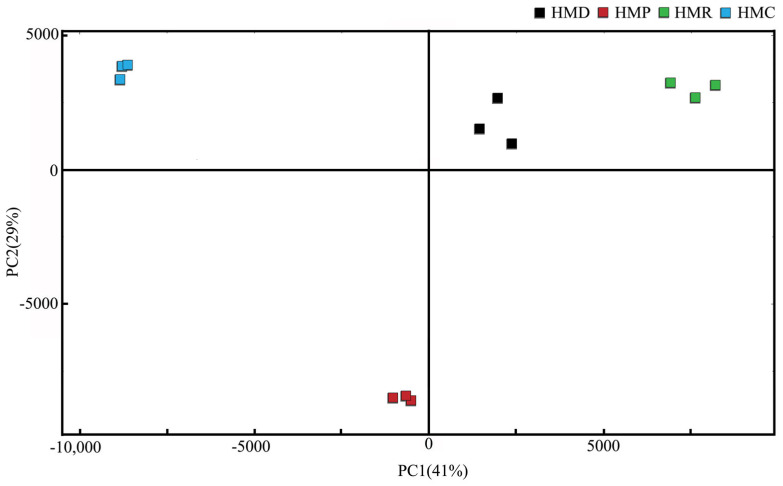
PCA analysis graph of volatile compounds in four hempseed fermented milk samples.

**Figure 5 foods-13-04076-f005:**
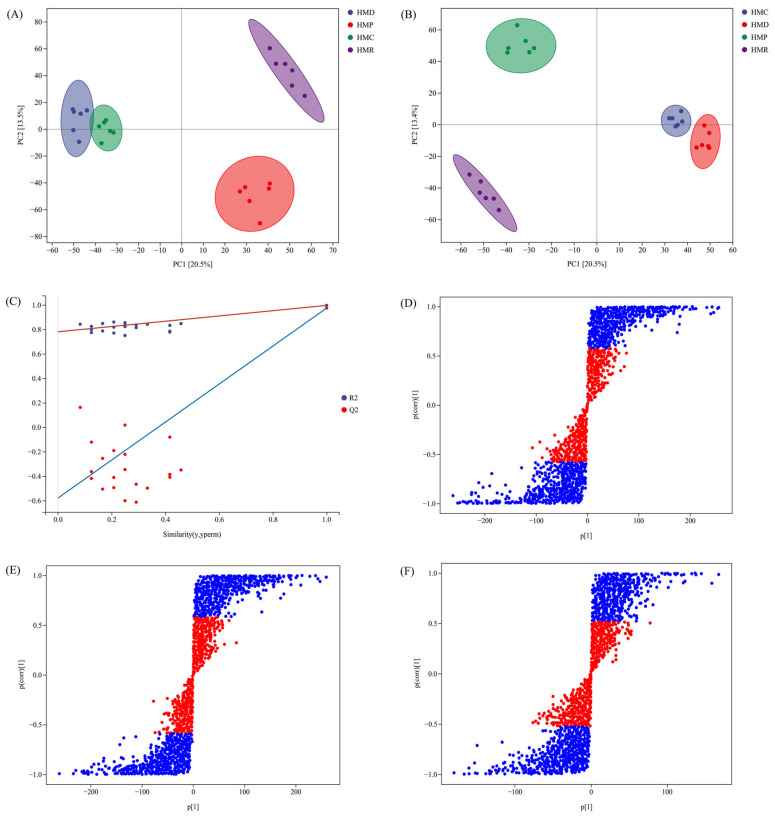
Multivariate statistical analysis of HMD, HMP, HMR, and HMC. (**A**) PCA score figure; (**B**) OPLS-DA score plot; (**C**) Cross-validation plot of OPLS-DA model; (**D**) S-plot of OPLS-DA showing the different metabolites between HMD and HMP; (**E**) S-plot of OPLS-DA showing the different metabolites between HMD and HMR; (**F**) S-plot of OPLS-DA showing the different metabolites between HMD and HMC.

**Figure 6 foods-13-04076-f006:**
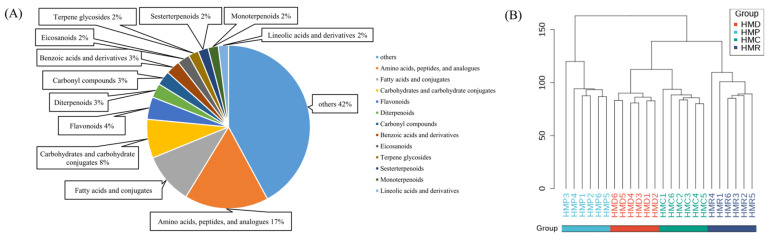
(**A**) Classification of 247 identified differential metabolites. (**B**) Hierarchical cluster of differential metabolites for HMD, HMP, HMR, and HMC.

**Figure 7 foods-13-04076-f007:**
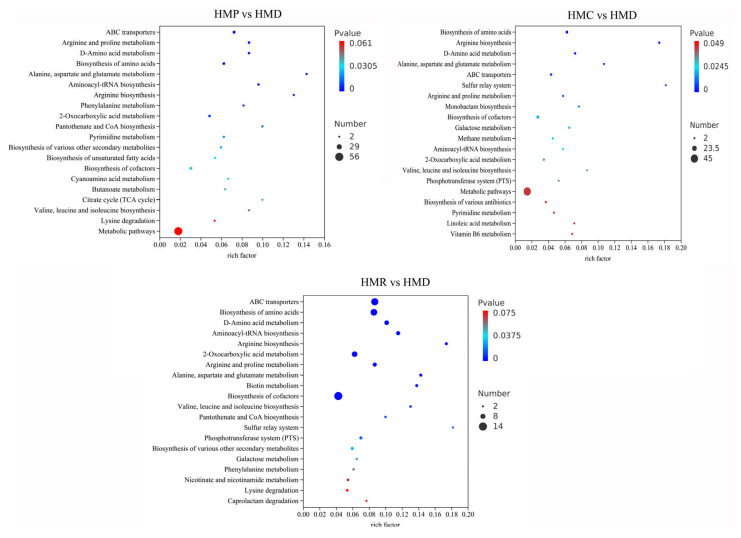
Overview of metabolic pathways significantly affected by fermentation with different probiotics in hempseed milk.

**Table 1 foods-13-04076-t001:** Texture of hempseed fermented milk. The combination of bacterial strains represented by different letter symbols are HMD: *L. bulgaricus* and *S. thermophilus*; HMR: single probiotic *L. rhamnosus* LGG and group HMD combination; HMC: single probiotic *L. paracei* LC-37 and group HMD combination; HMP: single probiotic *L. plantarum* XD117 and group HMD combination. The letters a, b indicated significant differences, with *p* < 0.05 indicating a significant difference and *p* > 0.05 indicating a non-significant difference.

	Hardness (gf)	Adhesiveness (gf*s)	Springiness	Cohesiveness	Gumminess (gf)	Chewiness (gf)	Resilience
HMD	10.859 ± 0.093 ^a^	−74.508 ± 2.019 ^a^	0.898 ± 0.001 ^a^	0.909 ± 0.003 ^b^	10.069 ± 0.150 ^a^	9.039 ± 0.089 ^a^	14,822.09 ± 546.36 ^a^
HMP	10.586 ± 0.119 ^b^	−70.337 ± 1.251 ^b^	0.906 ± 0.005 ^a^	0.928 ± 0.004 ^a^	9.872 ± 0.046 ^a^	9.040 ± 0.096 ^a^	14,006.46 ± 921.17 ^ab^
HMR	10.819 ± 0.22 ^ab^	−69.650 ± 1.264 ^b^	0.907 ± 0.002 ^a^	0.900 ± 0.006 ^b^	9.881 ± 0.132 ^a^	9.034 ± 0.166 ^a^	13,159.02 ± 971.12 ^b^
HMC	10.923 ± 0.052 ^a^	−70.730 ± 0.13 ^b^	0.899 ± 0.006 ^a^	0.912 ± 0.011 ^b^	9.892 ± 0.08 ^a^	9.033 ± 0.13 ^a^	14,582.47 ± 377.25 ^a^

**Table 2 foods-13-04076-t002:** Results of volatile significant flavor components of samples.

Compound	Relative Amount				Odor Description
	HMD	HMP	HMR	HMC	
Aldehydes					
Propanal	5.19 ± 0.64%	4.67 ± 0.12%	7.75 ± 0.09%	4.34 ± 0.07%	pungent, green, grassy
Acetaldehyde	3.84 ± 0.07%	4.28 ± 0.21%	5.00 ± 0.23%	2.93 ± 0.05%	green, slightly fruity
Hexanal	1.85 ± 0.64%	3.51 ± 0.07%	7.01 ± 0.2%	3.1 ± 0.17%	fresh, green, fat, fruity
(E)-2-Pentenal	1.06 ± 0.17%	1.76 ± 0.02%	1.22 ± 0.08%	2.08 ± 0.02%	potato, peas
3-Methylbutanal	0. 50 ± 0.06%	0.22 ± 0.01%	1.00 ± 0.06%	0. 42 ± 0.01%	chocolate, fat
alcohol					
Ethanol	15.74 ± 0.32%	11.1 ± 0.11%	11.23 ± 0.16%	11.69 ± 0.2%	aromaticity
1-Hexanol	3.62 ± 0.28%	5.93 ± 0.06%	2.59 ± 0.09%	3.31 ± 0.09%	fresh, fruity, wine, sweet, green
1-Propanol	3.32 ± 0.1%	1.44 ± 0.02%	1.58 ± 0.02%	1.84 ± 0.02%	alcohol, pungent
1-Penten-3-ol	2.16 ± 0.12%	2.44 ± 0.05%	1.99 ± 0.04%	2.97 ± 0.06%	ethereal, green, tropical fruity
1-Pentanol	1.23 ± 0.08%	1.72 ± 0.01%	1.49 ± 0.03%	2.24 ± 0.02%	balsamic
ketone					
2-Propanone	8.43 ± 0.47%	8.81 ± 0.14%	8.86 ± 0.13%	8.51 ± 0.14%	fresh, apple, pear
3-Hydroxy-2-butanone	4.87 ± 0.2%	6.49 ± 0.13%	3.87 ± 0.1%	4.4 ± 0.19%	fatty, oily, aldehyde, vegetable, cinnamon
2-Pentanone-D	3.70 ± 0.39%	4.07 ± 0.05%	3.71 ± 0.07%	5.16 ± 0.04%	acetone, fresh, sweet, fruity, winey,
2-Butanone	3.27 ± 0.18%	3.26 ± 0.05%	4.96 ± 0.08%	4.62 ± 0.17%	fruity, camphor
2,3-Butanedione	1.15 ± 0.09%	1.26 ± 0.02%	1.45 ± 0.07%	0.71 ± 0.02%	butter, popcorn, sweet taste, sour rice
2-Heptanone-M	1.22 ± 0.06%	0.78 ± 0.12%	1.64 ± 0.02%	0.18 ± 0.00%	pear, banana, fruity, slight medicinal fragrance
ester					
Ethyl acetate	9.24 ± 0.89%	6.88 ± 0.5%	8.55 ± 0.12%	9.34 ± 1.2%	fresh, fruity, sweet, grassy
carboxylic acid					
Acetic acid	16.29 ± 1.05%	17.43 ± 0.07%	12.32 ± 0.36%	19.22 ± 0.57%	spicy
sulfide					
Dimethyl sulfide	2.05 ± 0.09%	0.80 ± 0.05%	0.79 ± 0.15%	1.09 ± 0.05%	cabbage, sulfur, gasoline

**Table 3 foods-13-04076-t003:** Pearson correlation coefficients between chemical compositions and α-glucosidase inhibition activity and pancreatic lipase inhibition activity (*r* > 0.6).

	α-Glucosidase Inhibition Activity	Pancreatic Lipase Inhibition Activity
Flavonoids		
Panasenoside	0.987 ***	0.55
Genistein	0.906 ***	0.335
Rutin	0.912 ***	0.271
Amino acids and derivatives		
L-Arginine	0.747 **	0.938 ***
L-Tyrosine	0.495	0.712 **
L-Lysopine	0.736 **	0.176
N-methyl-L-glutamic Acid	0.306	0.762 **
N-Acetyl-L-phenylalanine	0.387	0.748 **
4-Guanidinobutanoic acid	0.249	0.665 *
Peptides		
N2-gamma-Glutamylglutamine	0.975 ***	0.442
Glutaminylmethionine	0.976 ***	0.477
Glycylleucine	0.253	0.732 **
Hydroxyprolyl-Lysine	0.438	0.646 *
Tryptophylhydroxyproline	0.209	0.803 **

Note: * presents a significant difference at *p* < 0.05; ** presents a very significant difference at *p* < 0.01; *** presents a very significant difference at *p* < 0.001.

## Data Availability

The original contributions presented in the study are included in the article/Appendix A, further inquiries can be directed to the corresponding author.

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
