# Peer review of "Effect of Different Probiotic Fermentations on the Quality of Plant-Based Hempseed Fermented Milk"

_foods, 2024, doi:10.3390/foods13244076_

Round 1

Reviewer 1 Report

Comments and Suggestions for Authors

Introduction:

Please contextualize the specific probiotic strains used, particularly regarding previous findings on their effects in other plant-based or fermented milk matrices.

Please highlighting why hempseed is especially suitable for probiotic fermentation or how it differs from other plant bases would add depth.

Materials and Methods

The methodology of the manuscript should have a control sample Including a non-fermented hemp milk control would help distinguish the effects solely due to fermentation.

Please explain clear how probiotic viability was maintained throughout the fermentation and storage periods would strengthen the rigor of the study. Additionally, explaining why 42°C was chosen as the fermentation temperature across all strains would be useful.

Structure the section more logically by grouping subsections into major areas, such as:

2.1. Microbial Culture and Preparation

2.2. Preparation of Hempseed Yogurt

2.3. Viability and Acid Production Analysis

2.4. Functional Analysis (α-Glucosidase and Pancreatic Lipase Inhibition)

2.5. Volatile and Non-Volatile Compound Analysis (GC-IMS and LC-MS)

2.6. Statistical Analysis

Section 2.2 why 42°C fermentation temperature? Did you made preliminary tests? for fermentation periods og 8 hours, with post-ripening at 12 hours?

Section 2.2 – I believe a table would be beneficial to present the formulations.

Section 2.3 - Please explain the “Three copies of laboratory counting.” Were the measurements made in triplicate ?

Results lacks detailed statistical explanations, please include detailed statistical annotations for multivariate analyses.

Why these functional characterizations: α-glucosidase and pancreatic lipase inhibition

Figure 2 and Figure 3A is nearly impossible to visualize

briefly discussing limitations, like the single base matrix (hempseed) and the focus on three probiotic strains, would provide a balanced perspective.

Conclusion

how these findings could guide probiotic selection in commercial plant-based milk products?

Author Response

Comment 1: In the first sentence of abstract the authors mentioned ”single probiotics”, did you mean single-strain probiotics? If so, it should be corrected in Conclusions as well and transferred to the plural, “…the effect of three different single-strain probiotics…” instead of “…the effect of a single probiotic…”.

Response:Agree. We have accordingly revised in Page 1, Line 25.

Comment 2: The whole Materials and methods section should be formulated differently, the passive should be used for describing the procedures. For example, “the method was used” instead of “use the method”; “the number of LAB was analyzed“ instead of “analyze the number of LAB”; “the yogurt was diluted” instead of “dilute the yogurt”; “a texture characteristic analyzer was used” instead of “use a texture characteristic analyzer”; “10g of fermented milk sample was weighed” instead of “Weigh 10g of fermented milk sample” etc. Authors should check the whole section and correct where needed.

Response:Agree. We have accordingly revised in Page 3-7, Line 110-209.

Comment 3: In 2.1. Materials: If CHR Hansen is a company that provided the yogurt starter, its location, city and country should be mentioned in parentheses (Hørsholm, Denmark for example). Same for all mentioned instruments and companies.

Response:Agree. We have accordingly revised in Page 4, Line 112.

Comment 4: In 2.1. Materials also: Since PBS is first mentioned here, the authors should indicate its full name, and the abbreviation in parentheses in this section rather than in 2.2.

Response:Agree. We have accordingly revised in Page 4, Line 117.

Comment 5: In 2.2. Hempseed yogurt preparation: What method was used to adjust and determine if the bacterial concentration is 1×109 CFU/mL? It should be described briefly.

Response:Agree. We have accordingly revised in Page 4, Line 121.

Comment 6: In 2.2. Hempseed yogurt preparation: The authors mentioned that the mixture was stirred and homogenized by using an experimental instrument. What kind of stirrer was used and what was the rotation speed (rpm-revolutions per minute), it should be mentioned.

Response:Agree. We have accordingly revised in Page 4, Line 125.

Comment 7: In 2.2. Hempseed yogurt preparation: The authors mentioned an ultra-low temperature refrigerator. The term deep-freezer or freezer instead of refrigerator is more suitable.

Response:Agree. We have accordingly revised in Page 4, Line 135.

Comment 8: In 2.3. Effect of different probiotics on viable count: “Analyze the LAB in fermented milk every 2 hours” should be corrected to “The number of LAB in fermented milk was analyzed every 2 hours”. Also, the dilution procedure was not formulated great. Maybe something like “serial dilutions were made by adding 1 mL of fermented milk to 9 mL of physiological saline (from 10−1 up to 10−5)” or similar would be better.

Response:Agree. We have accordingly revised in Page 4, Line 139-140.

Comment 9: In 2.3. Effect of different probiotics on viable count: Something similar to “the enumeration of LAB was done in triplicate” or “the enumeration of LAB was done in three repetitions” instead of “Three copies of laboratory counting” would make this part easier to follow.

Response:Agree. We have accordingly revised in Page 4, Line 142-143.

Comment 10: In 2.5. Texture the authors mentioned that the premeasurement, measurement, and post measurement rates were set to 0.5 mm/s, 0.2 mm/s, and 0.2 mm/s, respectively. The measurement and post measurement rates are the same. Please clarify if this is a mistake in the text.

Response:Yes, this parameter is correct.

Comment 11: GC-IMS Analysis: Same as point 4., since GC-IMS is first mentioned here, the authors should indicate its full name, and the abbreviation in parentheses. Other abbreviations should also be checked throughout the manuscript and the full name indicated where first mentioned.

Response:Agree. We have accordingly revised in Page 6, Line 176.

Comment 12: In the discussion authors explained in detail and connected their results and drew adequate conclusions. However, 3.1. section could be expanded and compared with results of other authors who dealt with similar problems.

Response:Agree. We have accordingly revised in Page 7, Line 218-231.

Comment 13: Also, in section 3.2. the results of cohesiveness, gumminess and chewiness from Table 1 are not commented on in the text. The authors should add a brief comment on these results.

Response:Agree. We have accordingly revised in Page 9, Line 271-272.

Comment 14: The text in Figure 3A and Figure 7 is small and hard to read, it should be enhanced for better visual representation.

Response:Agree. We have revised the picture to be better visual representation.

Comment 15: Somewhere in the text the authors put “yogurt” and other places “yoghurt”. It should be checked and corrected to be uniform and the same in the whole text.

Response:Agree. We have revised the misuse in the whole text.

Reviewer 2 Report

Comments and Suggestions for Authors

Dear editor, dear authors,

I have read carefully the paper entitled “Effect of different probiotic fermentations on the quality of plant-based hempseed fermented milk”. The paper covers the interesting topic of an alternative yogurt beverage and the addition of different probiotic strains to the fermentation process. Obtained beverages showed promising characteristics and different volatile and nonvolatile compounds present.

However, there are some issues throughout the text that need revision. My suggestion is that the paper should be made minor revisions to make it more readable and easier to follow.

SPECIFIC COMMENTS

  1. In the first sentence of abstract the authors mentioned ”single probiotics”, did you mean single-strain probiotics? If so, it should be corrected in Conclusions as well and transferred to the plural, “…the effect of three different single-strain probiotics…” instead of “…the effect of a single probiotic…”.
  2. The whole Materials and methods section should be formulated differently, the passive should be used for describing the procedures. For example, “the method was used” instead of “use the method”; “the number of LAB was analyzed“ instead of “analyze the number of LAB”; “the yogurt was diluted” instead of “dilute the yogurt”; “a texture characteristic analyzer was used” instead of “use a texture characteristic analyzer”; “10g of fermented milk sample was weighed” instead of “Weigh 10g of fermented milk sample” etc. Authors should check the whole section and correct where needed.
  3. In 2.1. Materials: If CHR Hansen is a company that provided the yogurt starter, its location, city and country should be mentioned in parentheses (Hørsholm, Denmark for example). Same for all mentioned instruments and companies.
  4. In 2.1. Materials also: Since PBS is first mentioned here, the authors should indicate its full name, and the abbreviation in parentheses in this section rather than in 2.2.
  5. In 2.2. Hempseed yogurt preparation: What method was used to adjust and determine if the bacterial concentration is 1×109 CFU/mL? It should be described briefly.
  6. In 2.2. Hempseed yogurt preparation: The authors mentioned that the mixture was stirred and homogenized by using an experimental instrument. What kind of stirrer was used and what was the rotation speed (rpm-revolutions per minute), it should be mentioned.
  7. In 2.2. Hempseed yogurt preparation: The authors mentioned an ultra-low temperature refrigerator. The term deep-freezer or freezer instead of refrigerator is more suitable.
  8. In 2.3. Effect of different probiotics on viable count: “Analyze the LAB in fermented milk every 2 hours” should be corrected to “The number of LAB in fermented milk was analyzed every 2 hours”. Also, the dilution procedure was not formulated great. Maybe something like “serial dilutions were made by adding 1 mL of fermented milk to 9 mL of physiological saline (from 10−1 up to 10−5)” or similar would be better.
  9.  In 2.3. Effect of different probiotics on viable count: Something similar to “the enumeration of LAB was done in triplicate” or “the enumeration of LAB was done in three repetitions” instead of “Three copies of laboratory counting” would make this part easier to follow.
  10. In 2.5. Texture the authors mentioned that the premeasurement, measurement, and post measurement rates were set to 0.5 mm/s, 0.2 mm/s, and 0.2 mm/s, respectively. The measurement and post measurement rates are the same. Please clarify if this is a mistake in the text.
  11. 2.8. GC-IMS Analysis: Same as point 4., since GC-IMS is first mentioned here, the authors should indicate its full name, and the abbreviation in parentheses. Other abbreviations should also be checked throughout the manuscript and the full name indicated where first mentioned.
  12. In the discussion authors explained in detail and connected their results and drew adequate conclusions. However, 3.1. section could be expanded and compared with results of other authors who dealt with similar problems. Also, in section 3.2. the results of cohesiveness, gumminess and chewiness from Table 1 are not commented on in the text. The authors should add a brief comment on these results.
  13. The text in Figure 3A and Figure 7 is small and hard to read, it should be enhanced for better visual representation.
  14. Somewhere in the text the authors put “yogurt” and other places “yoghurt”. It should be checked and corrected to be uniform and the same in the whole text.

Author Response

(The authors gave the same response as above.)

Reviewer 3 Report

Comments and Suggestions for Authors

The study  Effect of different probiotic fermentations on the quality of plant-based hempseed fermented milk determined the effects of probiotics Lactiplantibacillus plantarum XD117, Lacticaseibacillus paracasei LC-37, and Lacticaseibacillus rhamnosus LGG on the quality of fermented hemp seed milk. The results showed that adding probiotics had a small effect on titratable acidity, viable bacterial count, and textural properties. Still, they significantly increased the inhibition rate of α-glucosidase and pancreatic lipase. GC-IMS performed a general and validated analysis of the main flavor components, and a total of 19 major flavor compounds were found to be important in distinguishing different flavors. The results of untargeted metabolomic correlation analysis confirmed that 14 substances, including 3 flavonoids, 6 amino acids and their derivatives, and 5 short peptides, were positively correlated with the hypoglycemic and hypolipidemic activities of hemp seed fermented milk. A total of volatile flavor compounds, including aldehydes, alcohols, ketones, acids, and esters, were also identified, and the role mapping of different probiotic communities was provided. The results can be used for the development of hemp seed fermented milk with unique flavor, rich probiotic content, and significant functional characteristics.

Author Response

Comment 1: The study  Effect of different probiotic fermentations on the quality of plant-based hempseed fermented milk determined the effects of probiotics Lactiplantibacillus plantarum XD117, Lacticaseibacillus paracasei LC-37, and Lacticaseibacillus rhamnosus LGG on the quality of fermented hemp seed milk. The results showed that adding probiotics had a small effect on titratable acidity, viable bacterial count, and textural properties. Still, they significantly increased the inhibition rate of α-glucosidase and pancreatic lipase. GC-IMS performed a general and validated analysis of the main flavor components, and a total of 19 major flavor compounds were found to be important in distinguishing different flavors. The results of untargeted metabolomic correlation analysis confirmed that 14 substances, including 3 flavonoids, 6 amino acids and their derivatives, and 5 short peptides, were positively correlated with the hypoglycemic and hypolipidemic activities of hemp seed fermented milk. A total of volatile flavor compounds, including aldehydes, alcohols, ketones, acids, and esters, were also identified, and the role mapping of different probiotic communities was provided. The results can be used for the development of hemp seed fermented milk with unique flavor, rich probiotic content, and significant functional characteristics.

Response:Agree.

At the same time, please take into consideration the following requirements:

  1. Shuai Song & Haixuan Wang were employed by the company Jiangsu junyao Life Technology Development Co., What are their specific contribution to the manuscript?

Response:These two authors, as technicians, helped us establish and optimize the preparation process of hemp seed fermented milk, which is the basis for this study.

  1. What is the role of the company?

Response:The company is our cooperative enterprise, they have complete fermentation milk production equipment, can help us realize the production of hempseed fermentated milk

  1. Whether their involvement will have an impact on the fairness and objectivity of the outcome of the manuscript.

Response:Their contributions have been recognized by all of our authors, so we do not believe that impartiality will be affected.

  1. The remaining authors declare that the research was conducted in the absence of any commercial or financial relationships that could be construed as a potential conflict of interest”.

Response:We are very sorry for the misunderstanding, but we have no conflict of interest with this company.